# Superconductivity in High-Entropy Alloy System Containing Tb

**DOI:** 10.3390/ma18122747

**Published:** 2025-06-11

**Authors:** Piotr Sobota, Bartosz Rusin, Daniel Gnida, Adam Pikul, Rafał Idczak

**Affiliations:** 1Institute of Experimental Physics, University of Wrocław, pl. M. Borna 9, 50-204 Wrocław, Poland; piotr.sobota2@uwr.edu.pl (P.S.); bartosz.rusin@uwr.edu.pl (B.R.); 2Institute of Low Temperature and Structure Research, Polish Academy of Sciences, ul. Okólna 2, 50-422 Wrocław, Poland; d.gnida@intibs.pl (D.G.); a.pikul@intibs.pl (A.P.)

**Keywords:** high-entropy alloys, superconductivity, terbium

## Abstract

Superconducting alloy containing terbium (Tb) and its reference without the lanthanide were synthesized. X-ray diffraction, scanning electron microscopy, energy dispersive X-ray spectroscopy, specific heat, and magnetic measurements were used to investigate their structural and physical properties. Both alloys crystallized in body-centered cubic structure, and the presence of small amounts of Tb and Tb_2_O_3_ phases was detected. The critical temperature $T_{c}$ of alloys was in the 4.6–5.2 K range, and the upper critical field $\mu_{0}H_{c2}$ was 6.1–6.8 T. The comparison with the reference determined the effect of Tb on the alloy’s critical parameters and phase stability connected to the high-entropy alloys’ core effects. Overall, Tb addition did not have a beneficial effect on the superconducting properties of this alloy.

## 1. Introduction

High-entropy alloys are among the most studied materials in modern solid-state physics and materials science. They can be defined as solid solutions of five or more elements with at least 5 atomic percent each. Their name derives from the high value of entropy of mixing, which is responsible for stabilizing simple structures (such as body-centered cubic, face-centered cubic, and hexagonal close-packed) in these multicomponent materials. This is because increasing the amount of elements in a material usually leads to the precipitation of binary and ternary phases. In recent years, these materials have attracted considerable attention because of their remarkable mechanical properties; resistance to corrosion, including chemical corrosion; and stability over a wide temperature range [1,2,3,4]. These features are due to the so-called HEA core effects described by Yeh et al. [5,6].

Therefore, another consequence of these effects is the strong influence of internal chemical disorder and crystal lattice stresses on the superconducting properties of selected HEA alloys. Superconducting HEAs form a steadily growing family currently dominated by Nb-Ta matrices [7,8,9]. There are also titanium-rich systems [10,11] based on heavy transition metals [12] and partly ordered systems, [13] (which have some of the characteristics of HEAs). The formation of superconducting lanthanide-containing HEAs was first reported as early as 2015 [14], but to date there is very little information on superconductivity in such systems. Recently, a paper about a superconducting partly ordered system containing heavy lanthanides [15] was published, but there is no information about superconductivity in a system containing lanthanides which crystallize in a bcc structure.

In this study, we aim to explore whether the incorporation of terbium (Tb), a lanthanide with strong magnetic properties, into a high-entropy alloy matrix can influence its superconducting behavior. Specifically, we investigate whether Tb can increase the critical temperature or the upper critical field or modify the electron–phonon coupling strength. These properties are critical for potential applications of superconductors in high-field environments. Given the limited data on lanthanide-containing HEAs, especially those crystallizing in a bcc structure, our goal is to determine whether such systems can be synthesized and to evaluate the impact of Tb on their superconducting parameters.

Our starting point was the NbTa matrix, but in the course of our research we found that combinations of tantalum with lanthanides are impossible to obtain in an arc furnace due to differences in melting points and thermal conductivity (see Appendix A). Nevertheless, we were able to obtain a system with Tb, namely, (VNb)_0.67_(TiTbHf)_0.33_, and a corresponding lanthanide-free matrix with a formula (VNb)_0.753_(TiHf)_0.247_ to assess the effect of terbium on the superconducting properties of HEA.

## 2. Materials and Methods

Polycrystalline samples of (VNb)_0.67_(TiTbHf)_0.33_ and (VNb)_0.753_(TiHf)_0.247_ were synthesized by conventional arc melting in an atmosphere of Ti-treated Ar using elemental metal ingots of at least 99.9% purity. To remove residual, black terbium oxide from the surface of the pure terbium metal, the Tb ingot was polished using sandpaper. The melting process was repeated five times to ensure the homogeneity of the products. The mass loss was negligible (around 0.8% for Tb sample and below 0.2% for the reference). To enable structural and morphology studies, the samples were cut in half using a diamond circular saw and their surfaces polished.

The chemical composition and homogeneity of the samples were verified by energy-dispersive X-ray spectroscopy (EDXS) using a FESEM FEI Nova NanoSEM 230 scanning electron microscope equipped with an EDAX Genesis XM4 spectrometer on the surface of the cut and polished samples. The crystal structures of the as-obtained products were studied by powder X-ray diffraction (XRD) using a PANalytical X’pert Pro diffractometer with CuKα radiation. The experimental XRD patterns were analyzed using the Rietveld method implemented in the FULLPROF software [16].

The AC magnetic susceptibility of the alloys was studied in the temperature range of 1.8 K to 6 K and in magnetic fields up to 50 kOe using using a commercial Quantum Design MPMS-XL magnetometer. The heat capacity was measured from room temperature to 1.8 K and in fields up to 50 kOe using the Quantum Design PPMS and DynaCool platforms.

## 3. Results and Discussion

### 3.1. Crystal Structure

X-ray diffraction patterns of the obtained alloys are presented in Figure 1. The analysis of the experimental data by the Rietveld method showed that the (VNb)_0.753_(TiHf)_0.247_ alloy crystallizes in a body-centered cubic (bcc) structure; no traces of additional phases were detected. Reliability factors ($R_{p}$ = 2.81, $R_{\mathrm{wp}}$ = 4.23) commonly used to estimate the quality of the refinement demonstrated the good quality of this fit. The lattice parameter of the matrix was found to be *a* = 3.2568(1) Å.

The results of the XRD measurements indicate that there are three phases in the (VNb)_0.67_(TiTbHf)_0.33_ alloy: the expected structure of the bcc HEA, pure terbium precipitates with an amount of 2.9 wt.% estimated using Rietveld method, and a small amount (1.7 wt.%) of Tb_2_O_3_. The phases are listed in the order in which their Bragg reflections are presented (from top to bottom) in Figure 1b. The lattice parameter of the bcc phase was *a* = 3.2642(1) Å. The slight increase in the crystal lattice parameter is most likely related to the incorporation of Tb atoms. It is evident that some of the terbium atoms did not dissolve in the main phase and formed precipitates of the pure element. In addition, a passivation layer of terbium oxide must have formed on the surface of these precipitates (as the measurement was conducted on the surface of the cut and polished specimen and any residual oxides should, during the melting, migrate to the outer surface of the alloy ingot). Reliability factors $R_{p}$ = 1.35, $R_{\mathrm{wp}}$ = 1.76 demonstrated, again, the good quality of this fit.

### 3.2. Morphology and Chemical Composition

The analysis of the EDXS maps of the(VNb)_0.753_(TiHf)_0.247_ alloy, presented in Figure 2a, showed that while the sample lacks pure metal precipitates or areas that could indicate the presence of intermetallic compounds, there is a slight segregation into a niobium-rich HEA phase and a niobium-poor phase, with the titanium and vanadium content appearing to fluctuate less (see Table 1). The two phases form a mosaic pattern, and their crystal lattice constants must be very close to each other, even almost identical, since their separation is not visible in XRD measurements.

The micrographs of the (VNb)_0.67_(TiTbHf)_0.33_ alloy, presented in Figure 2b, show clear precipitates of terbium (and possibly terbium oxide in the same area, although oxygen was not mapped). Moreover, the main phase has a slightly different morphology from the matrix. First, the mosaic structure is only visible at higher magnification. Second, segregation is visible in the case of niobium. Local disturbances in the stoichiometry of the other elements are also present but they are not significant. However, this is not reflected in the XRD measurements, where additional phases (beyond the three mentioned in the previous section) are not visible.

### 3.3. Physical Properties

The results of the AC magnetic susceptibility measurements in various magnetic fields in the zero-field cooling (ZFC) regime are presented in Figure 3 and Figure 4. In both alloys, the real and imaginary parts of the susceptibility confirm the presence of bulk, type II superconductivity. The incorporation of the terbium atoms into the matrix causes the reduction in $T_{c}$ from about 5.2 K to 4.7 K. The critical temperatures were determined based on the position of the maximum in $\chi^{\prime \prime }\left(T\right)$ data.

Additionally, the degree of critical temperature change in relation to the induction of magnetic fields indicates that the sample containing terbium demonstrates a higher degree of decay in its superconductivity.

This happens because, as the temperature gets higher, magnetic flux lines and bulk shielding currents begin to penetrate the superconductor when the applied field exceeds the lower critical field. This destroys the ideal Meissner state and causes energy to be lost. When the flux lines and shielding currents fully penetrate the material, the losses reach a maximum. When the temperature reaches its critical value, the loss value goes to zero [17]. In the case of the(VNb)_0.753_(TiHf)_0.247_ sample, the superconducting state is still present in the field of 5 T, while the sample of (VNb)_0.67_(TiTbHf)_0.33_ becomes paramagnetic under the same conditions. No signals characteristic of any magnetic orders were detected. The main phase inhomogeneities seen in the SEM images must therefore have a negligible effect on the magnetic properties of the alloys.

Since the results of the magnetic measurements indicated the bulk nature of superconductivity, specific heat measurements were used for further analysis and to define the critical properties of the superconducting states. In the rest of this paper, we call the alloy (VNb)_0.753_(TiHf)_0.247_ ‘matrix’ and the (VNb)_0.67_(TiTbHf)_0.33_ alloy ‘HEA-Tb’.

The results of the measurements of the specific heat of the two alloys at a nominal zero field and temperatures from RT to 1.8 K are presented in Figure 5a,c. The overall shape of the experimental C_p_(T) curves indicate typical behavior of metal alloys. Namely, they follow the Dulong–Petit law and generally show no additional features except for the HEA-Tb alloy, where at about 220 K, an anomaly related to the ferromagnetic ordering of pure terbium can be observed [18]. The inserts of these two figures shows a low-temperature measurement range, revealing transitions to the superconducting state in both alloys studied. For the matrix, the transition has a typical λ shape and is quite sharp. $T_{c}$ calculated by the equal entropy method is 5.2(1) K. In the case of HEA-Tb, the transition is broadened. Before the transition, a slight lifting of the measurement points is also observed. In addition, an anomaly related to the ordering of antiferromagnetic terbium oxide Tb_2_O_3_
is observed (about 2 K [19]). The critical temperature, determined by the same method, was 4.6(1) K. Normalized specific heat jump $\Delta C_{p}/\gamma T_{c}$ was found to be equal to 1.49 (matrix) and 1.05 (HEA-Tb). The latter is significantly smaller than the value predicted by the BSC theory. This may be caused by the effect of terbium doping or by the effect of magnetic phases on the height of the measurement points.

At low temperatures the C_p_ (T) curve can be described as a sum of the electronic heat contribution (connected with γ coefficient) and the phonon heat contribution (with corresponding β and σ coefficients). The solid black curves represent the fits of Equation (Equation 1) to the experimental data.(1)$$C_{P}\left(T\right)=\gamma T+\beta T^{3}+\sigma T^{5},$$

The γ, β, and σ coefficients were found to be equal to 9.1(1) mJ K^−2^ mol^−1^, 0.079(4) mJ K^−4^ mol^−1^, and 5.6(4) $\times \;10^{-4}$ mJ K^−6^ mol^−1^ for the matrix and 11.1(2) mJ K^−2^ mol^−1^, 0.085(9) mJ K^−4^ mol^−1^, and 6.4(1) $\times \;10^{-4}$ mJ K^−6^ mol^−1^ for the HEA-Tb, accordingly.

Using the calculated value of the β coefficient and the Equation (Equation 2) (where R is a gas constant and r = 1), the Debye $\Theta_{D}$ temperatures were estimated for the two alloys as 291 K (matrix) and 284 K (HEA-Tb).(2)$$\beta =\frac{12}{5}rR\pi^{4}{\left(\frac{T}{\Theta_{D}}\right)}^{3},$$

The $T_{c}$ and $\Theta_{D}$ values were used to calculate electron–phonon coupling using McMillan’s Equation [20](3)$$\lambda_{\mathrm{el}-\mathrm{ph}}=\frac{1.04+\mu^{*}\ln\left(\frac{\Theta_{D}}{1.45T_{c}}\right)}{\left(1-0.62\mu^{*}\right)\ln\left(\frac{\Theta_{D}}{1.45T_{c}}\right)-1.04},$$
where $\mu^{*}$ is the Coulomb repulsion constant. Assuming $\mu^{*}$ = 0.125 (a value commonly used for *d*-electron systems), we obtained $\lambda_{\mathrm{el}-\mathrm{ph}}$ = 0.68 (matrix) and 0.62 (HEA-Tb), which places the alloys in weakly to intermediate coupling.

Using the relation:(4)$$\gamma =\frac{1}{3}\pi^{2}k_{B}^{2}N_{A}N\left(E_{F}\right)$$
where $k_{B}$ is the Boltzmann constant and $N_{A}$ is the Avogadro number. The experimental densities of states of conduction electrons at the Fermi level $N(E_{F})$ were estimated as 3.9 states eV^−1^ f.u.^−1^ (matrix) and 4.7 states eV^−1^ f.u.^−1^ (HEA-Tb). The densities of non-interacting electrons $N{\left(E_{F}\right)}^{*}$ were calculated as 2.3 states eV^−1^ f.u.^−1^ (matrix) and 2.9 states eV^−1^ f.u.^−1^ (HEA-Tb) using the following relation [21]:(5)$$N{\left(E_{F}\right)}^{*}=\frac{N(E_{F})}{1+\lambda_{\mathrm{el}-\mathrm{ph}}}.$$

Then, by subtracting the major phonon contribution to the specific heat $\beta T^{3}$ from the total heat $C_{p}$, it is possible to estimate its pure electron contribution to the specific heat $C_{\mathrm{el}}$ and—from its change with temperature—the superconducting energy gap $\Delta_{0}$. The inverse temperature dependence of the electron heat is shown as ln($C_{\mathrm{el}}/\gamma T_{c})\left(1/T\right)$ in the insets to Figure 5b,d. The solid black regression lines depict the least-squares fits of Equation [22]:(6)$$C_{\mathrm{BCS}}\left(T\right)=A\gamma T_{c}\exp\left(-\frac{\Delta_{0}}{k_{B}T}\right),$$
where *A* is a constant. By applying the above relation, we obtained $\Delta_{0}/k_{B}$ = 8.45 K (matrix) and 10.41 K (HEA-Tb), which after normalizations yielded $2\Delta_{0}/k_{B}T_{c}$ = 3.25 and 4.53, respectively. The latter deviates significantly from the BCS weak-coupling limit of 3.52, which is expected for a single, isotropic, superconducting gap. However, such elevated values are not uncommon in systems containing lanthanides and actinides [23]. This enhancement may point to strong electron–phonon coupling, consistent with the moderately high coupling constant ($\lambda_{\mathrm{el}-\mathrm{ph}}=0.62$) derived for HEA-Tb using McMillan’s formula.

The Pauli limiting fields $\mu_{0}H_{P}$ = 9.4 T (matrix) and 8.5 T (HEA-Tb) were then calculated from the following relation [24]:(7)$$\mu_{0}H_{P}=1.84T_{c}.$$

Also, using the calculated values of the energy gap, the thermodynamic critical fields 0.11 T (matrix) and 0.14 T (HEA-Tb) were calculated from the relation [25]:(8)$$\mu_{0}H_{c}\left(0\right)=\sqrt{\frac{3\gamma_{V}}{2\pi^{2}\mu_{0}}}\frac{\Delta_{0}}{k_{B}}.$$
where $\gamma_{V}$ is the volumetric Sommerfeld coefficient.

To estimate the values of the upper critical field $\mu_{0}H_{c2}\left(0\right)$, we studied the behavior of the anomalies in specific heat in different magnetic fields (see Figure 5b,d). The critical temperature values were used to construct the phase diagram (Figure 6). To make the estimation accurate, the Werthamer–Helfand–Hohenberg (WHH) [26,27,28] model for BCS superconductors in the dirty limit was used:(9)$$\ln\frac{1}{t}=\left(\frac{1}{2}+\frac{i\lambda_{\mathrm{SO}}}{4\gamma }\right)\psi \left(\frac{1}{2}+\frac{\bar{h}+\frac{1}{2}\lambda_{\mathrm{SO}}+i\gamma }{2t}\right)+\left(\frac{1}{2}-\frac{i\lambda_{\mathrm{SO}}}{4\gamma }\right)\psi \left(\frac{1}{2}+\frac{\bar{h}+\frac{1}{2}\lambda_{\mathrm{SO}}-i\gamma }{2t}\right)-\psi \left(\frac{1}{2}\right)$$
where $\alpha_{M}$ is the Maki constant, $\lambda_{\mathrm{SO}}$ is the spin–orbit scattering constant, $\gamma \equiv \sqrt{{\left(\alpha_{M}\bar{h}\right)}^{2}-{\left(\frac{1}{2}\lambda_{\mathrm{SO}}\right)}^{2}}$, $\bar{h}=\frac{4}{\pi^{2}}\frac{H_{c2}}{-dH_{c2}/dt}$, $t\;=\;\frac{T}{T_{C}}$, and ψ is a digamma function. The model includes the spin-paramagnetic effect and allowed the estimation of $\mu_{0}H_{c2}\left(0\right)$ (6.8 T for the matrix and 6.1 T for HEA-Tb), the orbital upper critical field $\mu_{0}H_{c2}^{\mathrm{orb}}$ (7.9(1) T for the matrix and 7.3(1) T for HEA-Tb), and $\alpha_{M}$ (1.18 for the matrix and 1.20 for HEA-Tb). Lowering the upper critical field by doping with lanthanide (terbium) therefore has the opposite effect to raising it when doped with scandium, a rare earth element, as presented in [29]. It is worth noting that XRD and SEM-EDXS analysis revealed the presence of secondary phases, specifically, Tb precipitates and Tb_2_O_3_. These phases likely contribute to the observed reduction in the superconducting volume fraction in the HEA-Tb sample. As these phases do not participate in the superconducting state, they may act as non-superconducting inclusions, thereby reducing the effective superconducting volume. While such inclusions can enhance flux pinning in some systems by acting as pinning centers, our results suggest that this effect is weak or even negligible. The lower upper critical field and broader superconducting transition in the HEA-Tb alloy support this interpretation.

In Figure 5d and in the inset of Figure 5c, it can be seen that there is some lifting of the measurement points before the main transition. Most likely, this is related to the Tb concentration gradient at the interphase between the alloy and pure Tb, similar to what was reported in the case of thorium-rich and thorium-poor phases in another HEA alloy superconductor [30].

The non-zero values of the spin–orbit scattering parameter $\lambda_{\mathrm{SO}}$ obtained from the WHH fits indicate that spin–orbit coupling plays a non-negligible role in the superconducting behavior of both alloys. This is particularly relevant in multicomponent systems, where the presence of heavy elements can enhance spin–orbit interactions. The extracted $\lambda_{\mathrm{SO}}$ values (1.2 for the matrix and 1.1 for HEA-Tb) suggest moderate spin–orbit scattering, which may influence the robustness of the superconducting state in the case of applied magnetic fields. These values are comparable to those reported for the (NbTi)_0.67_ (MoHfV)_0.33_ superconductor, which also exhibits comparable $\mu_{0}H_{c2}\left(0\right)=6.8\left(1\right)$ T [31]. In contrast, HEAs with significantly higher upper critical fields ($\mu_{0}H_{c2}\left(0\right)>12$ T), such as Ti_0.5_(ZrNbHfTa)_0.5_ and Ti_0.5_(VNbHfTa)_0.5_, show $\lambda_{\mathrm{SO}}$ values exceeding 3 [11]. This indicates a possible correlation between strong spin–orbit scattering and enhanced critical field in HEA superconductors.

In the next step, the Ginzburg–Landau parameters were estimated using the $\mu H_{c2}\left(0\right)$, $\mu_{0}H_{c2}^{\mathrm{orb}}$ and $\mu_{0}H_{c}$. The coherence lengths at 0 K $\xi_{\mathrm{GL}}\left(0\right)$ [32] are as follows:(10)$$\begin{array}{c} \xi_{\mathrm{GL}}\left(0\right)=\frac{\phi_{0}}{2\pi \mu_{0}H_{c2}^{\mathrm{orb}}\left(0\right)}, \end{array}$$
and were found to be 6.4 nm (matrix) and 6.7 nm (HEA-Tb). The values of the Ginzburg–Landau penetration depth at 0 K $\lambda_{\mathrm{GL}}$(0) were determined as follows:(11)$$\begin{array}{c} \lambda_{\mathrm{GL}}\left(0\right)=\sqrt{\frac{\phi_{0}\mu_{0}H_{c2}^{\mathrm{orb}}\left(0\right)}{4\pi \mu_{0}H_{c}{\left(0\right)}^{2}}}, \end{array}$$
and were found to be 320 nm (matrix) and 246 nm (HEA-Tb). Finally, the Ginzburg–Landau parameters $\kappa_{\mathrm{GL}}$ were defined as follows:(12)$$\kappa_{\mathrm{GL}}=\frac{\lambda_{\mathrm{GL}}}{\xi_{\mathrm{GL}}},$$
and were calculated as equal to 50 for matrix and 37 for HEA-Tb. Based on $\kappa_{\mathrm{GL}}$ and the thermodynamic critical field, the lower critical fields $\mu_{0}H_{c1}\left(0\right)$ were calculated using the following relation:(13)$$\mu_{0}H_{c1}\left(0\right)=\frac{\mu_{0}H_{c}\left(0\right)}{\sqrt{2}\kappa_{\mathrm{GL}}}ln\left(\kappa_{\mathrm{GL}}\right),$$
as equal to 0.0063 T (matrix) and 0.0097 T (HEA-Tb).

All the derived parameters describing the physical properties of both studied alloys are presented in Table 2.

## 4. Conclusions

The synthesis, structural, and physical properties of the high-entropy alloy system (VNb)_0.67_(TiTbHf)_0.33_ and the corresponding Tb-free matrix (VNb)_0.753_(TiHf)_0.247_ were reported and discussed. Refinement of the XRD data showed that both alloys crystallized in body-centered cubic structures, and in the case of the former alloy, some inhomogeneity of pure Tb and its oxide was detected. The results were confirmed by SEM-EDXS observations, which additionally showed some minor inhomogeneities in the main phases of both alloys. Measurements of magnetic susceptibility and specific heat have shown that terbium doping does not improve the critical parameters of the superconducting state of the bcc HEA alloy. However, there is no clear indication whether it is caused by the innate magnetic properties of terbium or an effect related to the electronic structure; therefore, further studies are required, possibly using other lanthanides and/or actinides.

## Figures and Tables

**Figure 1 materials-18-02747-f001:**
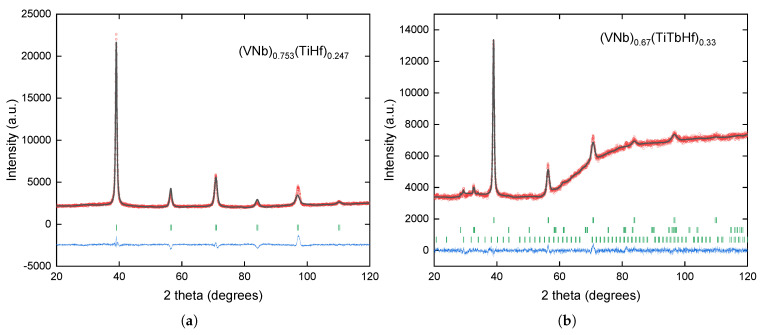
X-ray diffraction patterns and results of refinements for the (**a**) (VNb)_0.753_(TiHf)_0.247_ and (**b**) (VNb)_0.67_(TiTbHf)_0.33_ alloys. Red circles and gray lines represent the experimental points and the theoretical curves, respectively. Blue lines show a difference between the two, and green vertical dashes indicate positions of the Bragg reflections for the identified phases. In panel (**b**), the upper dashes correspond to bcc phase, the middle to pure Tb precipitates, and the bottom to Tb_2_O_3_.

**Figure 2 materials-18-02747-f002:**
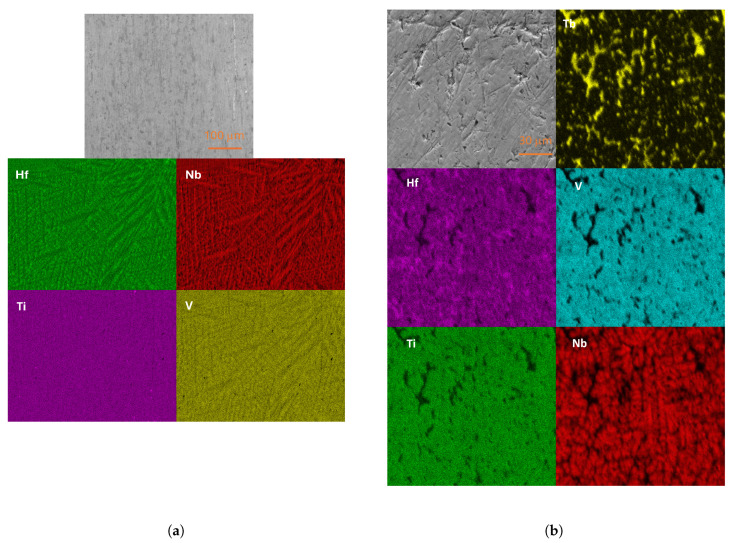
SEM micrographs with EDXS elemental mapping superimposed on the sample surface of the (**a**) (VNb)_0.753_(TiHf)_0.247_ and (**b**) (VNb)_0.67_(TiTbHf)_0.33_ alloys.

**Figure 3 materials-18-02747-f003:**
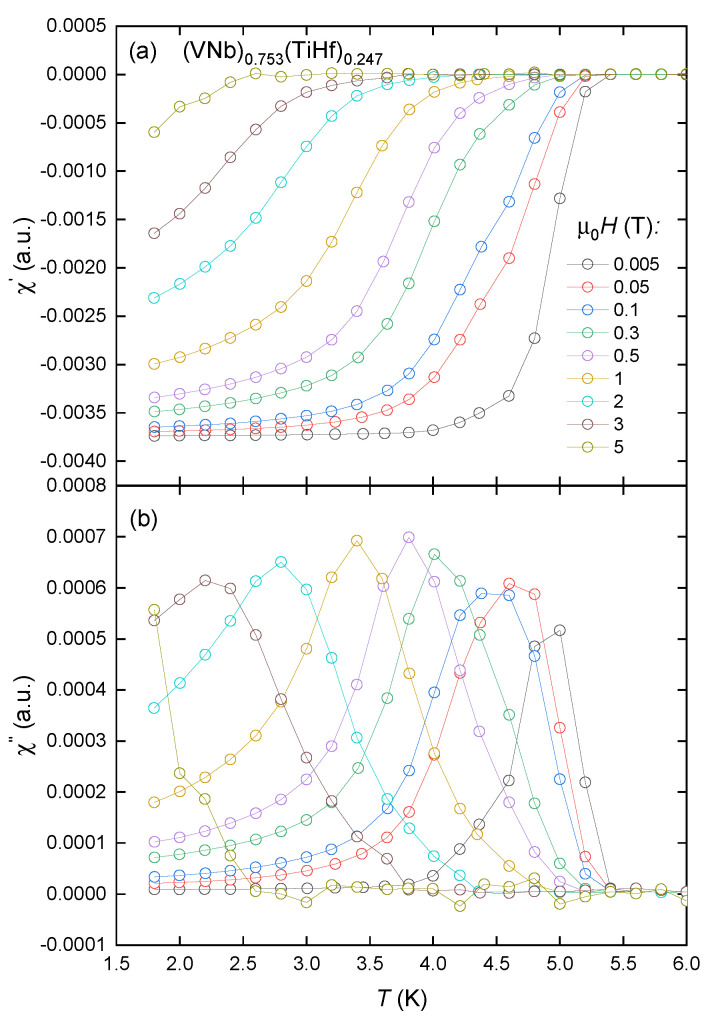
Real (**a**) and imaginary (**b**) parts of AC susceptibility measured for the (VNb)_0.753_(TiHf)_0.247_ alloy in the nominal applied magnetic fields $\mu_{0}H$. Hollow circles represent the measured points, and solid lines serve as a visual guide.

**Figure 4 materials-18-02747-f004:**
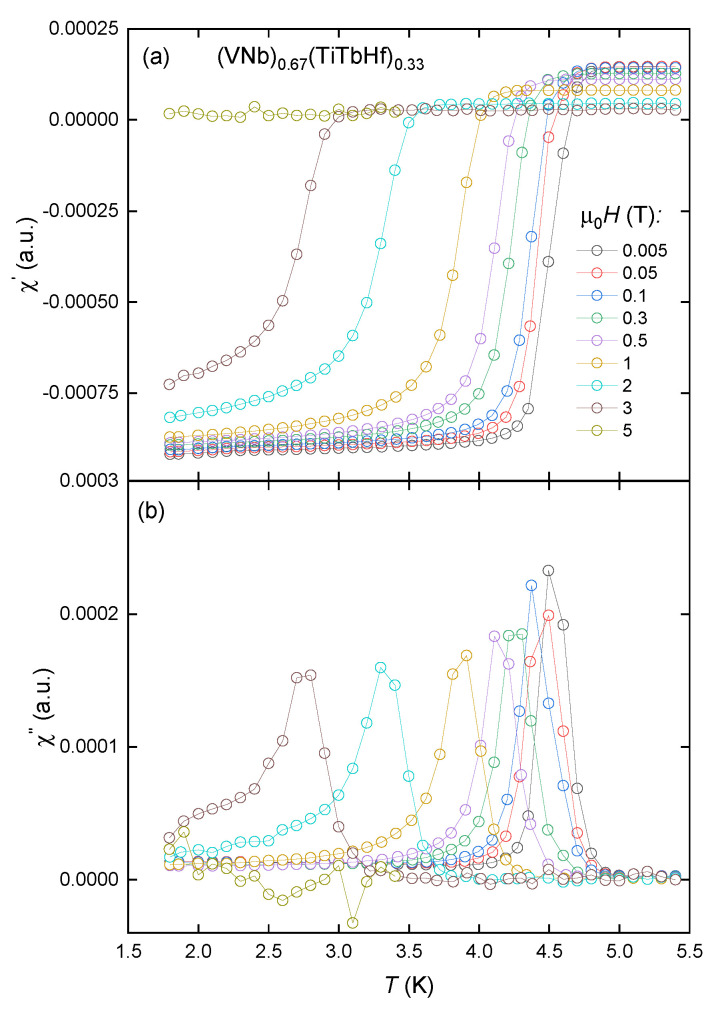
Real (**a**) and imaginary (**b**) parts of AC susceptibility measured for the (VNb)_0.67_(TiTbHf)_0.33_ alloy in the nominal applied magnetic fields $\mu_{0}H$. Hollow circles represent the measured points, and solid lines serve as a visual guide.

**Figure 5 materials-18-02747-f005:**
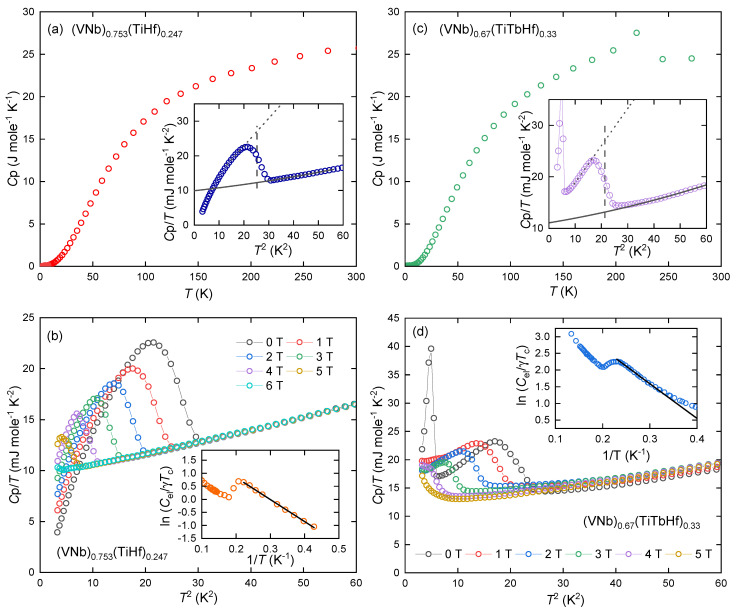
(**a**) Temperature variation of specific heat $C_{P}$ of (VNb)_0.753_(TiHf)_0.247_; the insert shows $C_{P}/$T($T^{2}$) measured in nominal zero magnetic field, and the solid curve depicts fit of Equation (Equation 1) to the experimental data together with dashed lines indicating the estimated normalized jump and $T_{c}$. (**b**) $C_{P}/T$ vs. $T^{2}$ measured in various external magnetic fields $\mu_{0}H$ up to 6 T for (VNb)_0.753_(TiHf)_0.247_ sample; the inset shows $\ln\left(C_{\mathrm{el}}/\gamma T_{c}\right)$ as a function of inverse temperature measured in a nominal zero field, and the straight solid line is a fit of Equation (Equation 6) to the experimental data. Panels (**c**,**d**) display analogous analysis for the (VNb)_0.67_(TiTbHf)_0.33_ alloy.

**Figure 6 materials-18-02747-f006:**
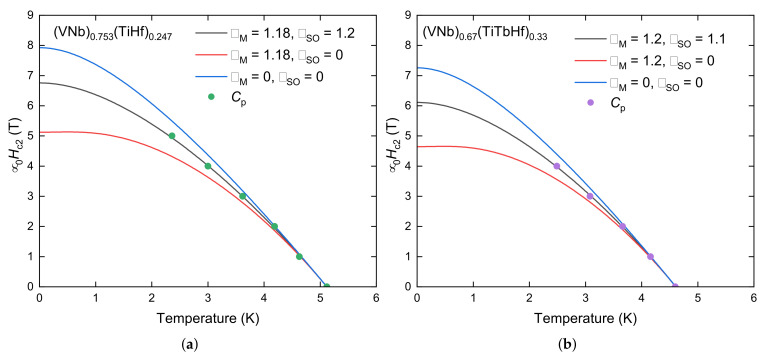
Upper critical fields of (**a**) (VNb)_0.753_(TiHf)_0.247_ and (**b**) (VNb)_0.67_(TiTbHf)_0.33_ as a function of temperature, derived from specific heat data. The lines depict the simulated WHH curves (Equation (Equation 9)).

**Table 1 materials-18-02747-t001:** Averaged compositions over several repeated measurements of similar areas of the main phase in the (VNb)_0.67_(TiTbHf)_0.33_ alloy and two representative areas present in the (VNb)_0.753_(TiHf)_0.247_.

	Element	Ti	V	Tb	Hf	Nb
(VNb)_0.67_(TiTbHf)_0.33_	at.%	12.13 ± 0.63	37.92 ± 0.89	2.73 ± 0.43	14.13 ± 1.26	33.09 ± 1.81
(VNb)_0.753_(TiHf)_0.247_: Nb-rich	at.%	12.95 ± 0.65	38.32 ± 0.79	—	12.53 ± 0.66	36.19 ± 0.83
(VNb)_0.753_(TiHf)_0.247_: Nb-poor	at.%	13.09 ± 0.70	39.30 ± 1.2	—	15.06 ± 1.0	32.56 ± 2.0

**Table 2 materials-18-02747-t002:** Experimentally determined basic characteristic parameters of normal and superconducting states in (VNb)_0.753_(TiHf)_0.247_ and (VNb)_0.67_(TiTbHf)_0.33_.

Parameter	(VNb)_0.753_(TiHf)_0.247_	(VNb)_0.67_(TiTbHf)_0.33_
$T_{c}$	5.2(1) K	4.6(1) K
γ	9.1(1) mJ K^−2^ mol^−1^	11.1(2) mJ K^−2^ mol^−1^
β	0.079(4) mJ K^−4^ mol^−1^	0.085(9) mJ K^−4^ mol^−1^
σ	5.6(4) $\times \;10^{-4}$ mJ K^−6^ mol^−1^	6.4(1) $\times \;10^{-4}$ mJ K^−6^ mol^−1^
$\Theta_{D}$	291(5) K	284(10) K
$\Delta C_{p}/\gamma T_{c}$	1.49	1.05
$\lambda_{\mathrm{el}-\mathrm{ph}}$	0.68(1)	0.62(1)
$N(E_{F})$	3.9(1) states eV^−1^ f.u.^−1^	4.7(1) states eV^−1^ f.u.^−1^
$N{\left(E_{F}\right)}^{*}$	2.3(1) states eV^−1^ f.u.^−1^	2.9(1) states eV^−1^ f.u.^−1^
$\mu_{0}H_{c}\left(0\right)$	0.11 T	0.14 T
$2\Delta_{0}/\left(k_{B}T_{c}\right)$	3.25	4.53
$\mu_{0}H_{c1}\left(0\right)$	0.0063 T	0.0097 T
$\mu_{0}H_{c2}\left(0\right)$	6.8 T	6.1 T
$\mu_{0}H_{c2}^{\mathrm{orb}}$	7.9(1) T	7.3(1) T
$\mu_{0}H_{P}$	9.4(1) T	8.5(1) T
$\alpha_{M}$	1.18	1.20
$\xi_{\mathrm{GL}}\left(0\right)$	6.4 nm	6.7 nm
$\lambda_{\mathrm{GL}}\left(0\right)$	320 nm	246 nm
$\kappa_{\mathrm{GL}}$	50	37

## Data Availability

The data presented in this study are openly available in the OSF repository at DOI 10.17605/OSF.IO/G4N6B.

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
