# Peer review of "Superconductivity in High-Entropy Alloy System Containing Tb"

_materials, 2025, doi:10.3390/ma18122747_

Round 1
Reviewer 1 Report
Comments and Suggestions for Authors
This study explores the synthesis and superconducting properties of a novel high-entropy alloy (HEA) system based on niobium and vanadium with the addition of terbium. The paper is well-written and structured, however, a minor revision is required:
- Elaborate on the significance of the elevated 2Δâ‚€/(kBTc) ratio (~4.53) in the Tb-doped sample: is this indicative of unconventional pairing, multiband effects, or strong coupling?
- While detecting Tb precipitates and Tb₂O₃, the study doesn't quantify how secondary phases affect superconducting volume fraction or flux pinning. Please add a brief discussion about this.
- Please clarify how the error bars were determined. Were these uncertainties derived from repeated measurements, fitting errors, or instrumental precision? If applicable, please state how many times each key measurement was repeated.
Author Response
1. Elaborate on the significance of the elevated 2Δâ‚€/(kBTc) ratio (~4.53) in the Tb-doped sample: is this indicative of unconventional pairing, multiband effects, or strong coupling?
This enhancement may point to strong electron-phonon coupling, consistent with the moderately high coupling constant derived for HEA-Tb using McMillan’s formula.
The appropriate comment was added in the text.
2. While detecting Tb precipitates and Tb₂O₃, the study doesn't quantify how secondary phases affect superconducting volume fraction or flux pinning. Please add a brief discussion about this.
Certainly, presence of additional phases, especially magnetic phases would decrease the superconducting volume fraction of the alloy. In this case they also have no positive effect on the upper critical field, contrary it is lower in alloy with Tb, thus effect on flux pinning is low.
Appropriate comments were added to the manuscript.
3. Please clarify how the error bars were determined. Were these uncertainties derived from repeated measurements, fitting errors, or instrumental precision? If applicable, please state how many times each key measurement was repeated.
Error bars presented in this work for EDXS measurements were calculated based on both the instrumental error of the detector and the statistical deviation of measurements of similar areas. The appropriate comment was added in the text. Other measurements were made without repetition.
Reviewer 2 Report
Comments and Suggestions for Authors
Specify in the abstract the most important results obtained. Highlight the importance of the use of Tb in the improvement of alloy properties.
The introduction describes the current situation of the alloys used and the importance of the proposed alloys, but it is not clear what properties are expected.
the results are well justified, tables are shown that could be more fully described and the graphs are adequate and clearly described and well justified.
Author Response
1. Specify in the abstract the most important results obtained. Highlight the importance of the use of Tb in the improvement of alloy properties.
We modified the abstract.
2. The introduction describes the current situation of the alloys used and the importance of the proposed alloys, but it is not clear what properties are expected.
In this study, we aim to explore whether the incorporation of terbium (Tb), a lanthanide with strong magnetic properties, into a high-entropy alloy matrix can influence superconducting behavior. Specifically, we investigate whether Tb can increase the critical temperature or the upper critical field, or modify the electron-phonon coupling strength. These properties are critical for potential applications of superconductors in high-field environments. Given the limited data on lanthanide-containing HEAs, especially those crystallizing in a bcc structure, our goal is to determine whether such systems can be synthesized and to evaluate the impact of Tb on their superconducting parameters.
An appropriate comment was added in the introduction.
Reviewer 3 Report
Comments and Suggestions for Authors
In their contribution "Superconductivity in high-entropy alloy system containing Tb", Sobota et al. report on the syntheses, structure determinations, magnetic properties and heat capacity measurements of two high entropy alloys. In the framework of their research, the authors conclude that both materials exhibit a metal-to-superconductor transition below 5 K. Although the research presented in that contribution agrees well with the scope of Materials, yet, there are certain issues, which must be fixed prior to a publication of this work:
- As terbium is sensitive against air and moisture, all preparations of samples containing terbium should always be accomplished under a dry inert gas atmosphere within a glove box. As the authors did not use such a glove box for the preparation of their samples, it could happen that the high-entropy alloys are also accompanied by oxide impurities, which could influence the results of all measurements. Therefore, the authors should clarify how an oxdiation of the employed metals was prevented during sample preparation. If an oxidation of the used metals could not be hindered, I strongly recommend to repeat the experiments by using Schlenk techniques as well as a glove box for sample preparation.
- The powder X-ray diffraction pattern shown in figure 1b clearly points to a technical issue concerning the measurement. Therefore, I strongly recommend to repeat the X-ray diffraction experiments as well as the Rietveld refinements in order to obtain an accurate determination of the composition.
- In the framework of the physical properties measurements, the authors should also accomplish temperature-dependent measurements of the electrical resistivity in order to provide further justification for the presence of a superconducting state. Furthermore, Tb2O3 should also exhibit a certain magnetic response. Did the authors identify any magnetic transition that could be related to the magnetic properties of Tb2O3? Furthemore, how could the magnetic properties of Tb2O3 influence the results of the magnetic measurements reported by the authors?
Author Response
1. As terbium is sensitive against air and moisture, all preparations of samples containing terbium should always be accomplished under a dry inert gas atmosphere within a glove box. As the authors did not use such a glove box for the preparation of their samples, it could happen that the high-entropy alloys are also accompanied by oxide impurities, which could influence the results of all measurements. Therefore, the authors should clarify how an oxdiation of the employed metals was prevented during sample preparation. If an oxidation of the used metals could not be hindered, I strongly recommend to repeat the experiments by using Schlenk techniques as well as a glove box for sample preparation.
Our synthesis was made under Ti-gettered argon atmosphere. Before synthesis surface of Tb was mechanically cleaned using sandpaper. The oxide detected by the XRD was formed most probably after the synthesis (during cutting and XRD measurement, it was exposed on the contact with air) on the surface of Tb precipitates. The amount of it is minuscule. Effects of this additional Tb and Tb2O3 phases were taken into account while preparing the manuscript.
2. The powder X-ray diffraction pattern shown in figure 1b clearly points to a technical issue concerning the measurement. Therefore, I strongly recommend to repeat the X-ray diffraction experiments as well as the Rietveld refinements in order to obtain an accurate determination of the composition.
We couldn't make a powdered sample so the XRD measurements were made on the cut surface of the sample; thus, this background elevation can be seen in higher angles. It did not affect the outcome of the measurement the exact compositional analysis based on the Rietveld refinement was added to the text.
3. In the framework of the physical properties measurements, the authors should also accomplish temperature-dependent measurements of the electrical resistivity in order to provide further justification for the presence of a superconducting state. Furthermore, Tb2O3 should also exhibit a certain magnetic response. Did the authors identify any magnetic transition that could be related to the magnetic properties of Tb2O3? Furthemore, how could the magnetic properties of Tb2O3 influence the results of the magnetic measurements reported by the authors?
We identified the terbium oxide magnetic response during the specific heat measurements (low temperature anomaly below 2K that shifts towards lower temperatures with increasing field) according to the reference [19] Hill, R.W. The specific heats of Tb2O3 and Tb4O7 between 0.5 and 22K. Journal of Physics C: Solid State Physics 1986, 19, 673. 262 https://doi.org/10.1088/0022-3719/19/5/007. As stated in the previous section oxide amount must be miniscule and it should be present only on the surface of the sample, which means that its magnetic response in comparison to the response of the main phase is more prevalent in small samples with high surface to bulk ratio like the one used in specific heat measurements (sample mass around 5-6 mg). In comparison, during the AC-susceptibility measurement sample more than 10x larger was used and there was no significant Tb2O3 anomaly detected, which further proves that Tb2O3 occurs mostly on the surface.
As almost all critical and thermodynamic properties of polycrystalline superconductors are derived from the specific heat and magnetic measurements, we don’t think that resistivity measurements are of any importance for the characterization of these materials' superconductive properties. Especially due to the oxide phase formation on the surface of the samples, which would disrupt the results. In short, we believe that this measurement in this case would require more time than the editorial deadline and bring no added value to the manuscript, as both AC and HC measurements extensively prove occurrence of superconductivity and provide data for all important calculations.
Reviewer 4 Report
Comments and Suggestions for Authors
High-entropy alloy (HEA) superconductors have attracted much attention since their relatively recent discovery due to their unique properties and potential for groundbreaking applications. This paper certainly adds to and expands our knowledge of this type of superconducting family.
Overall, I certainly will not reject that the authors have done a great amount of experimental work and have supported their findings to the best of their ability with theoretical justification.
However, I believe that the authors also need to do some more work on the manuscript and clarify a few important points:
1) The determination of the critical temperature was based on the diamagnetic response. Why didn't the authors (seem logical) to supplement the characterization of the samples with resistive measurements?
2) It would have been instructive for the authors to draw some sort of dashed line in the diamagnetic response figures to determine Tc and how exactly did they determine Tc?
3) As I understand it, in Figure 6, the theoretical description is only carried out on 4 experimental points in the middle and high temperature region? If so, it does not allow to speculate on the low-temperature behavior of the upper critical field dependence.
4) Thus, based on the numerical approximation within the WHH model and Figure 6, these superconductors demonstrate the presence of spin-orbital nature. Why do the authors do not discuss this in the text of the paper?
5) Since some HEA superconductors can demonstrate the multiband nature of the superconducting state, it would be reasonable to also try numerical modeling of the temperature dependence of the upper critical field within the framework of the two-band model (see, e.g., https://journals.aps.org/prb/abstract/10.1103/PhysRevB.67.184515 and Equation 34 there).
6) Technical remark. The coherence length and penetration depth correspond to T=0, so it is better to write \xi(0) and \lambda(0).
Author Response
1) The determination of the critical temperature was based on the diamagnetic response. Why didn't the authors (seem logical) to supplement the characterization of the samples with resistive measurements?
From our experience, resistivity measurements of polycrystalline superconducting samples do not provide significant new information, particularly when the sample is of a multiphase nature (with oxide) and when measurements of magnetic susceptibility and specific heat allow the superconducting state to be described in a consistent and comprehensive manner, which is the case here. This is not our default type of measurement in HEA alloys due to the difficult machining into the form required for this measurement.
2) It would have been instructive for the authors to draw some sort of dashed line in the diamagnetic response figures to determine Tc and how exactly did they determine Tc?
The Tc values were determined using the position of the maximum in imaginary part of AC-susceptibility measurement. Information about it was added to the text.
3) As I understand it, in Figure 6, the theoretical description is only carried out on 4 experimental points in the middle and high temperature region? If so, it does not allow to speculate on the low-temperature behavior of the upper critical field dependence.
The WHH model is intended to make it possible to predict the behaviour of the upper critical field as a function of temperature when measurements below 2 K are not possible which is our case. Whether the points are 4 or 5 does not affect the quality of the fit as they lie very well on the theoretical curve. Model is well embedded in the physics of BCS dirty limit superconductivity that it allows very little “movement” of the curve also in the low temperature region.
4) Thus, based on the numerical approximation within the WHH model and Figure 6, these superconductors demonstrate the presence of spin-orbital nature. Why do the authors do not discuss this in the text of the paper?
An appropriate comment was added in the manuscript:
The non-zero values of the spin–orbit scattering parameter λSO obtained from the WHH fits indicate that spin–orbit coupling plays a non-negligible role in the superconducting behavior of both alloys. This is particularly relevant in multicomponent systems, where the presence of heavy elements can enhance spin–orbit interactions. The extracted λSO values (1.2 for the matrix and 1.1 for HEA-Tb) suggest moderate spin–orbit scattering, which may influence the robustness of the superconducting state under applied magnetic fields. These values are comparable to that reported for the (NbTi)0.67(MoHfV)0.33} superconductor, which also exhibits comparable upper critical field (6.8(1) T). In contrast, HEAs with significantly higher upper critical fields (> 12$ T), such as Ti0.5(ZrNbHfTa)0.5} and Ti0.5(VNbHfTa)0.5}, show λSO values exceeding 3. This indicates a possible correlation between strong spin–orbit scattering and enhanced critical field in HEA superconductors.
5) Since some HEA superconductors can demonstrate the multiband nature of the superconducting state, it would be reasonable to also try numerical modeling of the temperature dependence of the upper critical field within the framework of the two-band model (see, e.g., https://journals.aps.org/prb/abstract/10.1103/PhysRevB.67.184515 and Equation 34 there).
Our current experimental data do not provide evidence supporting multiband superconductivity in the studied alloys. The temperature dependence of the upper critical field is well described by the single-band WHH model, and no anomalies or features indicative of multiband effects such as upward curvature were observed. Additionally, the electronic structure of the HEA matrix, dominated by d-band elements with relatively uniform density of states, suggests that a single-band approximation remains valid. Therefore, we did not pursue two-band modeling in this study, although we acknowledge that future spectroscopic investigations could further clarify this aspect.
6) Technical remark. The coherence length and penetration depth correspond to T=0, so it is better to write \xi(0) and \lambda(0).
Yes, we agree, its corrected now.
Round 2
Reviewer 3 Report
Comments and Suggestions for Authors
In the revised version of their contribution, the authors have addressed some of the reviewer's concerns; however, there are still certain issues which hinder a publication of this work:
The powder X-ray diffraction pattern shown in figure 1b is still of poor quality. Accordingly, it is not clear if the peaks of minor intensities are actually related to the materials as proposed by the authors. Furthernore, the authors provided the positions of the Bragg reflections, but did not mention which phase corresponds to the respective Bragg reflections.
I would assume that the first row of the green vertical dashes corresponds to the HEA. I fully agree that this phase has been obtained by the authors, and I think that terbium has been incorporated under consideration of the experimentally determined lattice parameters. Furthermore, I could imagine that the second row of the green vertical dashes corresponds to terbium pointing to terbium as a side-product; however, I am not sure if the third phase as represented by the third row of the green vertical lines is indeed present within the sample, as the quality of the pattern profile is quite poor.
So, if the third row of the green vertical lines corresponds to Tb2O3, it could also be possible that no Tb2O3 is actually present within the sample. The absence of any substantial amounts of Tb2O3 would account for the observed EDX and magnetic data, while the Tb2O3 at the surface could be just a result of passivation of the sample. On the other hand, terbium is also sensitive against air and moisture. Accordingly, the presence of the oxide could be just a consequence of slow surface oxidation of the terbium precipitates. If passivation of the surface is indeed a problem, it will certainly not make any sence to measure the temperature-dependent resistivity of the sample - a circumstance that should be discussed in detail in the main text.
Unfortnately, the authors' reply did not provide any information why it is not possible to obtain powders of that sample; however, it could be possible to obtain accurate powder patterns by using the Gandolfi technique. Furthermore, I also suggest to vary the acceleration voltage in the framework of the SEM micrographs and EDX elemental mapping. In doing so, it should be possible to analyze different layers of the sample. If the surface is oxidized, oxygene should be detected for the layers at as well as very close to the surface.
With regard to the authors' response: "In short, we believe that this measurement in this case would require more time than the editorial deadline and bring no added value to the manuscript, as both AC and HC measurements extensively prove occurrence of superconductivity and provide data for all important calculations."
It is not good if wrong results get published, as it will certainly disrupt the integrity of the authors, the reviewers as well as the journal. Therefore, I strongly suggest to go through my comments and fix all issues - even at the cost that it will not be possible to keep the editorial deadline. Maintaining scientific integrity is more important than every editorial deadline.
Furthermore, there are also several minor mistakes in the manuscript (see e.g all the references just showing question marks), which must be corrected.
Author Response
Please find the reply attached.

Reviewer 4 Report
Comments and Suggestions for Authors
I recommend this manuscript for publication